# Towards Efficient and Scalable Implementation of Differentially Private Deep Learning

## Abstract

Differentially private stochastic gradient descent (DP-SGD) is the standard algorithm for training machine learning models under differential privacy (DP). The most common DP-SGD privacy accountants rely on Poisson subsampling to ensure the theoretical DP guarantees. Implementing computationally efficient DP-SGD with Poisson subsampling is not trivial, which leads to many implementations that ignore this requirement. We quantify the computational cost of training deep learning models under differential privacy by benchmarking efficient methods with the correct Poisson subsampling requirement. We find that using the naive implementation DP-SGD with Opacus in PyTorch has a throughput between 2.6 and 8 times lower than that of SGD. However, efficient gradient clipping implementations like Ghost Clipping can roughly halve this cost. We propose alternative computationally efficient ways of implementing DP-SGD with JAX that use Poisson subsampling and performs comparably with efficient clipping optimizations based on PyTorch. We highlight important implementation considerations with JAX. Finally, we study the scaling behavior using up to 80 GPUs and find that DP-SGD scales better than SGD.

## 1. Introduction

Machine learning (ML) models' training data can be vulnerable to extraction (Balle et al., 2022; Carlini et al., 2021). Differential Privacy (DP) (Dwork et al., 2006) is the gold standard for formalizing the privacy leakage of training data in ML and mitigating the risk of privacy attacks on the training data. DP is deployed in many applications that involve sensitive data (Abowd, 2018; Cormode et al., 2018).

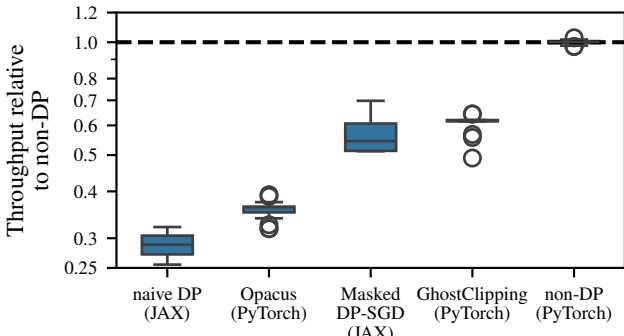

*Figure 1.* Relative throughput (FP32) to the respective non private baseline (higher is better) on NVIDIA A100. For each optimization method and each model size, we divide its throughput with the non-private counterpart. Throughput is the number of processed instances per second. In this benchmark we distinguish between precision modes. They are available on both frameworks and significantly improve the throughput for the different DP-SGD implementations.

The established algorithm for integrating DP into the training pipeline of deep learning models is DP stochastic gradient descent (DP-SGD) (Rajkumar & Agarwal, 2012; Song et al., 2013; Abadi et al., 2016), which is the DP adaptation of SGD (see also Alg. 1). DP-SGD has two major drawbacks in comparison to SGD: higher computational cost and loss in utility. DP-SGD requires more memory and is computationally more expensive due to the per-example clipping. The utility in comparison to non-DP training drops, but this can be mitigated to some extent by using larger batch sizes (Räisä et al., 2024) and training longer (Ponomareva et al., 2023) which further increase the computational cost.

Standard DP privacy accountants assume so-called Poisson subsampling, where each example is selected independently at each iteration with a fixed probability. This implies that different minibatches will be of different sizes, making efficient implementation more difficult. As a result, many existing implementations forego proper implementation of Poisson subsampling. Recent research (Lebeda et al., 2024; Chua et al., 2024a;b; Annamalai et al., 2024) shows that such implementations may have significantly weaker privacy guarantees than claimed under the Poisson subsampling assumption.

[1]Anonymous Institution, Anonymous City, Anonymous Region, Anonymous Country. Correspondence to: Anonymous Author <anon.email@domain.com>.

Preliminary work. Under review by the International Conference on Machine Learning (ICML). Do not distribute.

**List of contributions** In this work we conduct an extensive empirical study on the computational efficiency of DP-SGD using Poisson sub-sampling, focusing on fine-tuning a wide range of large image classification models. Our findings can be applied to any other large models trained or fine-tuned with DP-SGD. Our main contributions are as follows:

1. We re-implement all DP-SGD methods with Poisson subsampling that is fully DP and share the source code.
2. We propose a JAX implementation relying on proper Poisson sampling that is in comparison to a naive JAX implementation not prone to re-compilation and outperforms its throughput by two times (See Sec. 3).
3. We find that non-optimized training with DP-SGD costs per-epoch between 2.6 and 3.2 times more than non-private training for ViT and 4 to 8 times for ResNets (See Sec. 5). We identify the reasons that lead to the higher computational cost of DP-SGD using profiling.
4. We benchmark different strategies that can drastically reduce this cost: (i) More efficient gradient clipping implementations of DP-SGD (See Fig. 1 and Sec. 6.1). (ii) Lower Precision with TF32 (See Sec. 6.2).
5. We scale up the training to 80 GPUs and find that DP-SGD scales better than non-private training (See Sec. 8).

## 2. Background

This section will explain the main DP-SGD algorithm and optimizations to alleviate its computational cost.

### 2.1. DP-SGD Algorithm

Alg. 1 is the original DP-SGD algorithm, with virtual batching, as proposed by Abadi et al. (2016).

**Virtual Batching** distinguishes between logical and physical batches. Logical batches are divided into multiple physical batches to enable optimizer steps with many samples without running out of memory. For instance, we typically sample logical batch sizes of $L = 25000$ while the memory fits $< 300$ samples at a time. Implementing DP-SGD with virtual batching Alg. 1 does not modify the privacy accounting. The amount of noise added is the same and does not affect the model utility (Ponomareva et al., 2023).

**Poisson subsampling** Interestingly, Bu et al. (2022) and Bu et al. (2023) never mention Poisson subsampling in their works of Mix Ghost clipping and Book Keeping. Furthermore, Bu et al. (2022) claims a speed-up of $\times 1.7$ against other algorithms with a fixed batch size, which would affect the privacy accounting method. The same happens in practice for JAX implementations (De et al., 2022), where sampling is done by shuffling the dataset and using each sample once per epoch. While this makes efficient implementation easier, it does not use the correct Poisson subsampling assumed by privacy accounting methods. Therefore,

---

**Algorithm 1** Virtual Batching DP-SGD

---

**Input:** Training data points $\{x_1, \ldots, x_N\}$, loss function $\mathcal{L}(\theta) = \frac{1}{N} \sum_i \mathcal{L}(\theta, x_i)$
**Parameters:** learning rate $\eta_t$, noise scale $\sigma$, gradient norm bound $C$, number of steps $T$, approximate logical batch size $L$, physical batch size $p$.
**for** $t \in [T]$ **do**
     $B \leftarrow \{x_{j_1}, \ldots, x_{j_m}\}$ sample with rate $L/N$.
     $P \leftarrow \{B_1, \ldots, B_k\}$ split $B$ into physical batches of size $p$.
     $\theta_{acc} \leftarrow \mathbf{0}$
     **for** $s \in [P]$ **do**
         For each $i \in s$ compute $\mathbf{g}_t(x_i) \leftarrow \nabla_{\theta_t} \mathcal{L}(\theta_t, x_i)$
         $\overline{\mathbf{g}}_t(x_i) \leftarrow \mathbf{g}_t / \max \left( 1, \frac{\|\mathbf{g}_t(x_i)\|_2}{C} \right)$ {**Clip gradient**}
         $\theta_{acc} \leftarrow \theta_{acc} + \sum_i \overline{\mathbf{g}}_t(x_i)$ {**Accumulate gradient**}
     **end for**
     $\widetilde{\mathbf{g}}_t \leftarrow \frac{1}{|L|}(\theta_{acc} + \mathcal{N}(0, \sigma^2 C^2 \mathbf{I}))$ {**Add noise**}
     $\theta_{t+1} \leftarrow \theta_t - \eta_t \widetilde{\mathbf{g}}_t$ {**Step**}
**end for**
**Return** Learned parameters $\theta_T$ and the privacy cost from a privacy accountant.

---

the implementation might have significantly weaker privacy properties than claimed (Lebeda et al., 2024; Chua et al., 2024a;b; Annamalai et al., 2024). All our experiments are based on Poisson subsampling which is compliant with the commonly used privacy accounting.

### 2.2. DP-SGD Gradient Clipping Optimizations

We benchmark five types of clipping methods. Table 1 shows which clipping optimizations we are benchmarking against the library or framework that implements it.

**Ghost clipping** computes the loss gradient norm after the backpropagation step and then reweights the loss to update the clipped gradients. Its main contribution is memory savings at the cost of adding another backward pass (Li et al., 2022).

**Mixed Ghost clipping** (Bu et al., 2022) is a method that builds on-top of Ghost clipping. It implements the ghost clipping technique for convolutional layers. Its main contribution is that the algorithm will decide when to clip the gradients using per-example or ghost. This difference matters because the ghost clipping is less efficient when the layer's input dimensions are too high-dimensional. E.g., for ResNets, each clipping method will be applied for half of the layers. The first layers will be clipped using the the per-example and then ghost clipping in the bottom layers. As the model goes deeper, the feature size decreases, and the number of channels increases, prioritizing ghost clipping (Bu et al., 2022).

*Table 1.* Benchmarked DP-SGD frameworks and libraries. Note that Opacus Ghost Clipping is in development.

| | | PYTORCH | | | JAX | |
| CLIPPING MODE | NATIVE | OPACUS (YOUSEFPOUR ET AL., 2021) | PRIVATEVISION (PV) (BU ET AL., 2022) | FASTDP (BK) (BU ET AL., 2023) | NATIVE | OURS |
|---|---|---|---|---|---|---|
| NON-PRIVATE | ✓ | | | | ✓ | |
| PER-EXAMPLE | | ✓ | | | ✓ | |
| GHOST CLIPPING (LI ET AL., 2022) | | (✓) | ✓ | ✓ | | |
| MIX GHOST (BU ET AL., 2022) | | | ✓ | ✓ | | |
| MIX OPT (BU ET AL., 2023) | | | | ✓ | | |
| MASKED DP-SGD (OURS, SEC. 3) | | | | | | ✓ |

**Book Keeping** (Bu et al., 2023) uses all the previous techniques but requires only one backpropagation pass without explicitly calculating the per-example gradients. It avoids the second pass that ghost clipping does by reusing the intermediate results of the output gradients to calculate the sum of the clipped gradients and the clipping factor. Book Keeping can also be implemented together with the Mix Optimization, which does the same evaluation as the mix ghost clipping, but also determines whether doing a second backward pass is more efficient.

## 3. Avoiding Re-compilation in JAX

Using JAX for DP-SGD introduces complexities, particularly around Poisson subsampling which is crucial for privacy accounting. Implementing Poisson subsampling results in variable logical batch sizes that lead to variability in the size of the last physical batch which require JIT to recompile, leading to graph retracing which is costly and contributes to execution run variability (Chua et al., 2024a).

**Masked DP-SGD** We propose an algorithm, called masked DP-SGD, that overcomes the issue of recompilation at the cost of computing slightly more gradients than the naive implementation while at the same time using proper Poisson subsampling and therefore ensuring the correct privacy budget. We execute the following sub-steps at every iteration and highlight the differences to the naive implementation (steps 2 and 4) (See also Alg. A1):

1. We sample a logical size using Poisson sampling.
2. *We round up the number of samples for which we compute per-sample gradients so that it is divisible by the physical batch size without remainder.*
3. We compute the per-sample gradients.
4. *We mask out gradients so that the per-sample gradients used for the update are the actual Poisson subsampled ones, ensuring compliance with the Poisson subsampling accounting.*

**Extra computational cost** In step 2, we round the logical batch size up to the closest larger integer divisible by the physical batch size to avoid recompiling. Hence, for any sampled logical batch size $X$, the difference between $X$

and the upscaled batch size will be in $\{0, \ldots, p-1\}$ for a physical batch size $p$. Denoting the excess batch size with $\Delta_p(X)$ and the upscaled batch size with $X_+$, we can write

$$\mathbb{E}[X_+] = \mathbb{E}[X + \Delta_p(X)]. \tag{1}$$

Now, we can form a simple upper bound for the expected relative increase of batch size given that $\mathbb{E}[X] = L$ as

$$\mathbb{E}[X_+]/\mathbb{E}[X] \leq 1 + (p-1)/L. \tag{2}$$

When working large number of samples and non-negligible sampling probabilities, the excess cost due to upscaling the batch size will be modest for feasible physical batch sizes. For example, in our experiments the expected batch size of the Poisson subsampling was $L = 25\,000$, whereas the physical batch sizes extended up to $p = 64$. The expected relative increase in computed gradients would be $0.252\%$.

A recent work by Chua et al. (2024b) proposed an alternative implementation for JAX compilable implementation of Poisson subsampled DP-SGD. In their approach the logical batch sizes are sampled from a truncated Binomial distribution. In App. D we show that for our settings the number of additionally computed gradients is signficantly smaller with our method.

## 4. Experiment Overview

**PyTorch implementations** We benchmark a native PyTorch (Ansel et al., 2024) implementation with PyTorch-based libraries Opacus (Yousefpour et al., 2021) (details on gradsampling in App. A.3), PrivateVision (PV) (Bu et al., 2022), and FastDP (BK) (Bu et al., 2023), see Table 1. At submission time ghost clipping in Opacus was still undergoing changes and was unstable in our experiments.

**JAX implementations** We benchmark two JAX implementations. Our method Masked DP-SGD and a native JAX (Bradbury et al., 2018) implementation that clips the per-sample gradients with Optax (DeepMind et al., 2020) without utilizing any further optimization. This naive implementation in JAX is prone to recompilation due to changing tensor sizes caused by the Poisson subsampling.

**Implementation of Poisson sampling** Opacus samples the logical batches using Poisson sampling and then divides them into physical batches using their `BatchMemoryManager` class. The other PyTorch implementations considered in our experiments do not support virtual batching out-of-the-box. To make a fair comparison between all methods, we implemented Poisson subsampling in the same way as Opacus for all frameworks and adapted the `BatchMemoryManager` to support them. Thus, all experiments are seeded to ensure the same logical batch sizes.

**Metrics** We compare the throughput, defined as how many samples can be processed per second during training, and the maximum physical batch size that can fit in memory.

**Dataset** We benchmark with the CIFAR100 (Krizhevsky & Hinton, 2009) resized to 224x224.

**Models** We train two families of models: Vision Transformer (ViT) (Dosovitskiy et al., 2021) and ResNet (Kolesnikov et al., 2020) (See Table 2). Both are pre-trained on ImageNet-21k (Russakovsky et al., 2015).

*Table 2.* Number of parameters (millions) for used models.

| Vision Transformer (ViT) | | ResNet | |
|---|---|---|---|
| Type | # Params | Type | # Params |
| Tiny | 5.7 M | $50{\times}1$ | 23.7 M |
| Small | 22.1 M | $101{\times}1$ | 42.7 M |
| Base | 86.6 M | $50{\times}3$ | 211.8 M |
| Large | 304.3 M | $101{\times}3$ | 382.4 M |
| Huge | 630.8 M | $152{\times}4$ | 929.2 M |

**Parameterization** While parameter-efficient fine-tuning of some parts of the model has been shown to be effective under DP (Yu et al., 2022; Tobaben et al., 2023), our work focuses on the computational efficiency of DP-SGD and thus we consider the worst-case scenario of fine-tuning all parameters of the model. Furthermore, any training from scratch requires training all parameters.

**Hyperparameters** We train for four optimization steps with a sampling rate of 0.5 (expected batch size of 25000), which allows us to quickly test the experiments with a realistic high batch size (Ponomareva et al., 2023; Räisä et al., 2024). We do not focus on finding the best possible utility, which requires training for many more epochs (See Table A2 for the accuracy after training for four steps).

**Environment specifications** We use two GPU architectures: NVIDIA V100 (32 GB VRAM) and A100 (40 GB VRAM) with identical Python environments. Each node contains four GPUs. We use 16 CPU workers for data loading. In the distributed case of more than one GPU, only one worked per device is used.

**Source code** We provide the code for reproducing the experiments in the supplementary material and will publish the code in an open repository after acceptance of the paper.

# 5. What is the Computational Cost of DP in Deep Learning

We quantify the computational cost of deploying DP training by comparing the throughputs and maximum physical batch sizes between the non-private training with PyTorch and private training with Opacus, the most widely used DP-SGD implementation. Additionally, we identify the reasons for the higher computational cost of DP-SGD through profiling.

## 5.1. Throughput and Maximum Batch Size Comparison

We compare relative throughput (Fig. 2) and the maximum physical batch size (Fig. 3) between DP-SGD (Opacus) and non-private training with PyTorch. The main metric of interest is the throughput as it quantifies the training speed, but the maximum physical batch size becomes important when training models that are too large to fit even one example at a time. For both metrics, DP-SGD becomes more expensive with larger models, but the detailed trends differ.

**Vision Transformer** The throughput difference between Opacus and the non-private baseline with PyTorch (see Fig. 2(a)) grows steadily as a function of model size, which is interesting considering how big the relative difference in the maximum physical batch size (Fig. 3(a)) is: the throughput ranges from a relative difference of $\times 2.6$ for the smallest model to $\times 3.17$ for the largest model while the maximum physical batch size has a relative difference of around $\times 4$ for the smallest model and $\times 11$ for the largest model.

**ResNets** As depicted in Fig. 2(b), we observe a more irregular throughput and relative slowdown for the ResNets models size as their size grows. The contrast in Fig. 2 between ViT and ResNet models is due to the architecture and types of layers. The parameter space grows as the width factor (see Table 2) for the ResNets, so the $\times 3$ makes the neural network wider by a factor of three. Based on our results, the width of the layers affects throughput much more than the depth of the network. ResNet models with the same width and different depths exhibit comparable throughput, but increasing the width will make the model in the private setting much slower and reduce the maximum batch size significantly.

**How much does finding the maximum physical batch size matter?** In Fig. A.3 in the Appendix, we display the relative throughput as a percentage by dividing the throughput at a particular physical batch size by the maximum achievable throughput. We see that as the physical batch size increases, the throughput will grow as expected, but there is no significant further improvement at some point. Practitioners may estimate the optimal batch size based on available

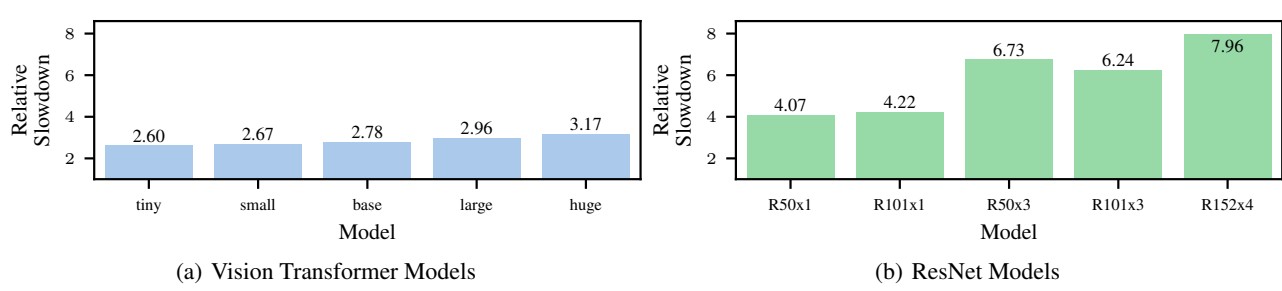

(a) Vision Transformer Models    (b) ResNet Models

*Figure 2.* Relative slowdown in mean throughputs between Opacus per-example clipping and the non-private baseline (A100 GPU). The relative slowdown is calculated as the ratio of private-throughput to non-private-throughput. A lower value indicates a better performance, closer to 1 indicates that Opacus is as fast as non-private training. This highlights the computational cost associated with private training.

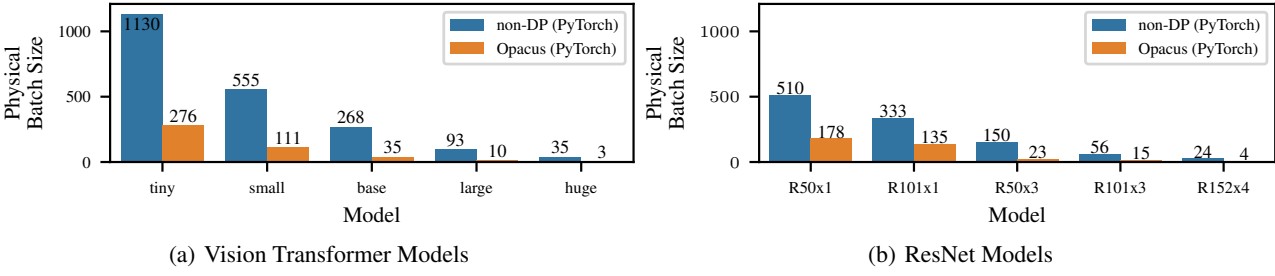

(a) Vision Transformer Models    (b) ResNet Models

*Figure 3.* Maximum achievable physical batch size by the different model sizes on A100 GPU (40 GB) before they reach Out Of Memory (OOM) Error. The model sizes grow from left to right (Refer to Table 2 for number of parameters).

*Table 3.* Average processing time in milliseconds for each section of the algorithm. We are comparing the non-private and Opacus clipping on A100, with the same physical batch size. We profile the time using NVIDIA Nsight Systems. All the measurements include the synchronization time, which is needed for the profiling, but adds additional time that is not part of the normal execution.

| SECTION | NON-DP (PYTORCH) | OPACUS (PYTORCH) |
|---|---|---|
| FORWARD | 81.14 | 101.53 |
| BACKWARD | 163.85 | 681.48 |
| CLIP & ACCUMULATE | 0 | 26.76 |
| OPTIMIZER STEP | 38.17 | 99.65 |

memory and performance trade-offs. Using the maximum physical batch size is not crucial, but a large enough value is sufficient. Typically, the throughput of smaller batches is limited by data loading speeds, but computation becomes the limiting factor as batch size increases.

### 5.2. Reasons for the Increase in Computational Cost

Giving a detailed breakdown of low-level operations associated with DP is challenging. However, using GPU profiling tool NVIDIA Nsight System, we can identify three aspects which constitute the majority of DP overheads. Firstly,

due to its larger memory footprint, DP-SGD is limited to consume smaller physical batches than its non-private counterpart. This results in a larger amount of smaller low-level kernel calls, which leads to slightly lower utilization of the GPU. Even the kernel launch overheads can become a notable factor for a slowdown at very small batch sizes. Secondly, the computation of per-example gradients introduces significant overhead in the backward pass as it cannot be parallelized as in batched gradient computation. This is the most prominent cause of the total overhead. Finally, an additional DP-optimizer step that clips and accumulates the per example gradients, which is not present in the non-DP algorithm, must be taken after each physical batch (see Table 3).

## 6. Decreasing the Computational Cost

This section analyzes the different strategies for training with DP-SGD more efficiently. We evaluate both algorithmic and hardware optimizations and their combinations.

### 6.1. Efficient Gradient Clipping Algorithms

First, we evaluate the more efficient gradient clipping implementations that have been described in Sec. 2.2 using the Vision Transformer base model. We chose it as our bench-

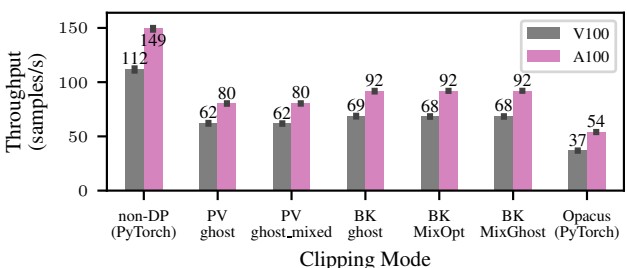

Figure 4. Throughput using the maximum batch size for each clipping algorithm. It compares the executions for both V100 and A100, for the ViT Base model.

Table 4. Maximum physical batch size reachable for each clipping method and GPU using for the ViT base model.

| CLIPPING MODE | V100 (32GB) | A100 (40GB) |
|---|---|---|
| NON PRIVATE BASELINE | 216 | 268 |
| PER-EXAMPLE (OPACUS) | 28 | 35 |
| GHOST (PRIVATE VISION) | 203 | 257 |
| MIX GHOST (PRIVATE VISION) | 203 | 257 |
| BK GHOST (FASTDP) | 189 | 209 |
| BK MIX GHOST (FASTDP) | 189 | 209 |
| BK MIX OPT (FASTDP) | 189 | 209 |

mark model because the middle model size is large enough to evaluate the advantages of the optimized gradient clipping algorithms but does not require excessive amount of time to train. The non-Opacus implementations do not support the ResNet due to their custom weight standardization layer.

**Throughput Comparison** Fig. 4 displays the throughput for each clipping algorithm for each tested GPU. Moving from a V100 to an A100 GPU increased the throughput by ×1.3 times on average over all clipping methods. The one that benefited the most is the per-example clipping by Opacus with a ×1.46 improvement in throughput. This is because of Opacus-specific optimizations. Their implementation is optimized to vectorize the virtual batches and get the most out of the processing unit to compensate for the per-example clipping. We base our virtual batching module on Opacus, which may have further contributed to the advantage seen for Opacus. The other implementations showed benefits similar to those of non-private training. For both GPUs, the clipping optimizations consistently maintained their relative throughput difference to their non private baseline. Private Vision gets closer to the non-private baseline physical batch size, but Book Keeping is closer to its throughput with a smaller physical batch size (see Fig. 6).

Without sacrificing utility (see Table A2), these optimizations offer an alternative to the original per-example clipping algorithm. Although Book Keeping has a slightly better throughput, the margin is narrow, making Private Vision and FastDP viable options as ghost clipping implementations. The difference between the two algorithms is the second backward pass over the neural network. The Book Keeping trick avoids this second backward pass, resulting in higher throughput at a small memory cost.

Mixed ghost clipping does not yield any improvement because it determines whether to apply ghost or per-example clipping, based on the size of the inputs and the parameter space. For large dimensions, ghost clipping will be more expensive (Bu et al., 2022). In ViT models, the dimensions change less than in a convolutional network. Therefore, despite continually evaluating which method to apply, it

consistently defaults to ghost clipping. Conversely, mix optimization applied to a ResNet model should outperform ghost clipping since it is optimized for convolutional layers. This could not be tested on ResNet models due to incompatibilities with Private Vision and FastDP, preventing an assessment of mixed optimization methods.

**Maximum physical batch size** Table 4 compares the maximum physical batch size for both available GPUs. The maximum physical batch size is larger for the optimizations of DP-SGD than for Opacus because they do not require per-example gradients. Consequently, these optimizations enable training much larger models without exhausting memory. The maximum physical batch size using the Private Vision library is the closest to the non-private baseline. Generally, the methods are consistent within implementations, with Private Vision and FastDP achieving the same maximum physical batch size regardless of the clipping mode. As expected, the A100 consistently attains higher maximum physical batch sizes than the V100 due to its larger VRAM.

### 6.2. Lower Precision

We consider using lower precision to speed up computation. We evaluate the use of TensorFloat-32 (TF32) for training. TF32 has 10 bits for precision, with eight range bits, giving it the same range but less precision than 32-bit single precision floats (FP32) (Kharya, 2020). Using lower precision can have benefits exactly where DP training struggles: it requires less memory, uses fewer bits to represent the data, and its operations are optimized for GPU, making them much faster (NVIDIA, 2023). It is special math mode introduced for the A100 GPU and unavailable for the V100, so we compared training on the A100 with and without TF32.

**Experimental results** In Fig. 5, we display the relative difference between mean throughput using TF32 and FP32. For non-private training, throughput increases with model size. For private training throughput increases for the smaller models, but it goes down again as the model size grows after the base size. Models that are too small do not gain much from TF32, and the larger ones have too small

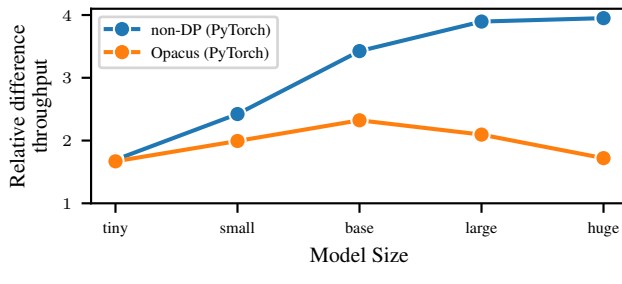

*Figure 5.* Relative difference in mean throughput between TF32 and FP32 Training for ViT Models.

maximum physical batch size to benefit (See detailed discussion of this in App. C). Regarding the memory advantages by TF32, we could not see an improvement. The maximum physical batch size is unaffected by the precision.

**Concerns regarding TF32 under DP** There are two concerns with using lower precision in DP deep learning: its effects on utility and privacy. Lower precision may affect utility, as it is less precise. We did not find a significant decay in the accuracy of the models compared to the models with FP32; it differs by decimal points at the $\times 10^{-6}$ precision (See Table A2). Regarding privacy, all floating point implementations provide imperfect implementations of real-valued mechanisms, that might introduce additional privacy vulnerabilities (Mironov, 2012). Lower precision may exacerbate this issue. Discrete mechanisms (e.g. Canonne et al., 2020; Agarwal et al., 2021) avoid these theoretical challenges, but are often less convenient and may reduce utility, especially in low precision settings. The efficiency of different discrete mechanisms in TF32 is an interesting topic of further research.

## 7. Comparison of JAX Implementations

We compare the performance of a naive non-private JAX, a naive JAX, and our proposed masked DP-SGD method with all other DP-SGD frameworks (all based on PyTorch). The utility is the same as in PyTorch (See Table A2). To provide a fair comparison, we implemented non-private and native DP JAX training using the same virtual batching as PyTorch. Note that JAX defaults to TF32 when available and FP32 needs to be explicitly forced.

**Throughput comparison (FP32)** In Fig. 6 (left), we compare the throughput using FP32. The naive DP-SGD JAX is the slowest implementation due to the JAX recompilation. Our proposed method masked DP-SGD outperforms Opacus and nearly matches the performance of PV Ghost Clipping despite not utilizing any optimizations regarding clipping. The masked DP-SGD exhibits higher throughput compared to other JAX implementations. This is primarily because the entire logical batch is accommodated in CPU memory, allowing it to be split into static sizes. Consequently, the

compilation time is elevated for the first logical batch; however, subsequent iterations benefit from increased speed as recompilation is unnecessary. In Fig. 6 (middle) we compare the throughput using TF32 and for this precision the results indicate that masked DP-SGD performs comparably to Opacus in terms of throughput. However, our method performs better on regimes with fewer samples (Fig. A.2) and allows for a larger physical batch size (Fig. 6 right).

**Compilation** The compilation time must be taken into account, given that the DP-SGD implementations in PyTorch do not compile. We measure it as the duration to process the first batch, since the execution times for each batch show that the first batch takes much more time than the others, including the compilation time (see Fig. A.4). The compilation time increases with batch size. For the private model, the compiled function is more complex than the non-private counterpart. It includes expanding the dimensions and clipping the gradients, while the non-private directly computes the gradient of the whole mini-batch.

Although compiling PyTorch is possible, we did not observe any significant speed improvements. Compiling the non-private model yielded minimal speed-up, but ultimately even lower when accounting for the compilation. PyTorch also recompiles after a batch size change, but reverts to predefined CUDA optimized operations. In the private setting, the compilation does not recognize Opacus hooks and continues the execution without compiling them (See Fig. A.5). Leveraging the same kernels to support the private hooks and avoid the compilation would require massive engineering work of writing special kernels for each specific private case. On the other hand, JAX will compile the JIT functions in XLA, but it does not fall back to the kernels, making it more generalizable (Subramani et al., 2021).

## 8. Distributed Training

We will look at another angle to train deep learning with DP faster: increasing the computational resources enough to decrease the training time. This is relevant when training cost or resource constraints are less important than the time to train a new model.

We utilize V100 GPUs on HPC nodes that have 4 GPUs per node. The other experimental setting is identical to the one in Sec. 5. Results for utilizing up to 24 A100 GPUs can be found in Fig. A.7 in the Appendix. We focus on comparing the scaling behavior between the non-private baseline that uses PyTorch and the DP-SGD implementation using Opacus. Both frameworks provide mature tooling for distributed training.

Fig. 7 shows the throughput increase as a function of number of GPUs. The throughput does not grow linearly and changes from the ideal linear scaling after using more than

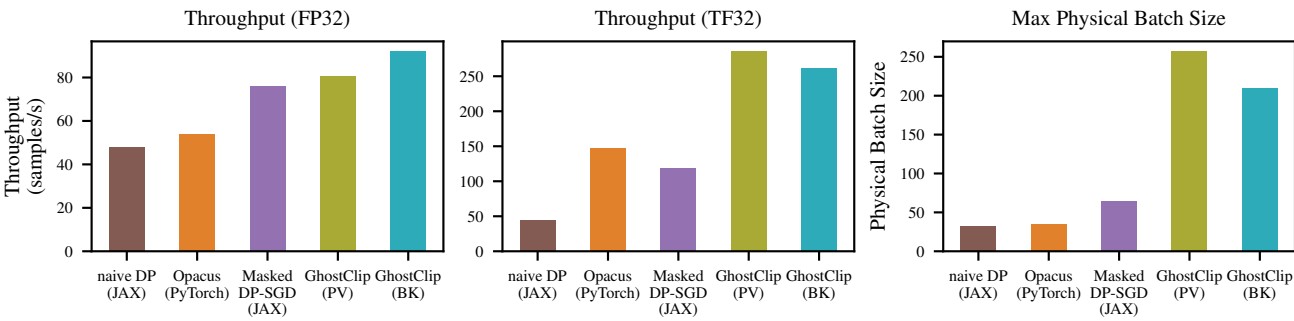

*Figure 6.* Throughput comparison across precision modes, for the ViT Base model, trained on A100 GPU. Using a lower precision should increase memory capacity and speed-up the sample processing. The results confirm that throughput is enhanced with lower precision. However, the physical batch size remained constant across precision modes.

*Table 5.* A summary of the lessons learnt. The relative throughput/max physical batch size is in comparison to PyTorch non-DP (higher is better) on A100. For each optimization method and each model size, we divide it with the non-private counterpart.

| Method | Relative to non-DP (PyTorch FP32) | | Supports | Compilation | | Section |
|---|---|---|---|---|---|---|
| | Throughput (↑) | Max Physical Batch Size (↑) | all layers | Initial | Re- | |
| **Opacus** | 0.31-0.39 | 0.08-0.24 | ✓ | - | - | Sec. 5 |
| **Efficient Gradient Clipping** | 0.49-0.54 | 0.88-0.95 | ✗ | - | - | Sec. 6.1 |
| **Native JAX** | 0.39-0.59 | 0.23-0.43 | ✓ | ✓ | ✓ | Sec. 7 |
| **Masked DP-SGD (ours)** | 0.51-0.69 | 0.11-0.23 | ✓ | ✓ | ✗ | Sec. 7 |
| **Masked DP-SGD + TF32** | 0.79-1.33 | 0.11-0.23 | ✓ | ✓ | ✗ | Sec. 7 |
| **Low Precision (Opacus+TF32)** | 0.54-0.84 | 0.08-0.24 | ✓ | - | - | Sec. 6.2 |

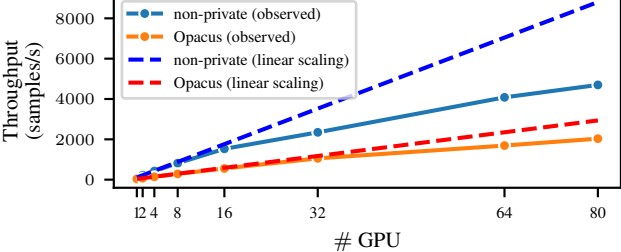

*Figure 7.* Comparison between the throughput by scaling the number of GPUs for the non-private and Opacus training with the ViT base model on V100 GPUs. The dashed line is the ideal growth.

one node (i.e. more than 4 GPUs). While the communication inside the node is fast, the communication between nodes will always be slower. The bottleneck is the network bandwidth, and it prevents the throughput from scaling linearly. Notably, it affects the non-private training baseline much more, while the private scales close to optimal up to 32 GPUs. For the 80 GPUs, the private training achieves 69.2% of the ideal linear speed-up, and the non-private training only achieves 53.3%. Private training scales better because it is slower and only sometimes saturates the network with updates. If we use Amdalh's law to compare the parallelism percentage for each case, we can see that in the private case, we achieve a 99.5% parallelism compared to a 98.9% in the non-private case (See Fig. A.8).

## 9. Conclusion

We summarize the lessons learnt in Table 5. While DP-SGD is significantly more costly than non-private training, we identified feasible speed-ups that are often easy to apply but have some drawbacks. These are: (i) More efficient implementations of DP-SGD which additionally decrease the memory footprint (allowing for training larger models). However, these implementations are not as mature as Opacus and do not support all neural network layers (yet). (ii) JAX lacks a comprehensive DP-SGD implementation like PyTorch and exhibits greater variability in execution times. Although JAX processes samples faster than PyTorch, it loses the advantage through frequent re-compilations when utilizing proper Poisson sampling. We present an efficient DP-SGD implementation with JAX called Masked DP-SGD. It leverages JAX advantages in compilation and efficient sample processing, while adhering to Poisson subsampling requirements for correct privacy accounting. By avoiding frequent recompilation, we mitigate execution time variability and enhances efficient performance. (iii) Lower Precision using TF32 which increases throughput but the implications on the theoretical guarantees of DP-SGD need to be explored in future work. Finally, we found that distributed computing using DP-SGD scales better than non-private training and allows for fast training of models.

## Impact Statement

This paper presents work whose goal is to advance the field of Machine Learning. There are many potential societal consequences of our work, none which we feel must be specifically highlighted here.

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

## A. Training Details

### A.1. Models

- Vision Transformer (ViT) (Dosovitskiy et al., 2021). Taken from `https://huggingface.co/timm/vit_base_patch16_224.orig_in21k`

- Big Transfer ResNet (Kolesnikov et al., 2020). Taken from `https://github.com/google-research/big_transfer`

### A.2. Hyperparameters

We use the hyperparameters obtained on request from Tobaben et al. (2023). The hyperparameters for both models are in Table A1. Even though model utility is not the main objective in this work, in the non-private case, the learning rate is suboptimal. By changing it to 0.00027 we see an accuracy improvement, therefore the one we are using.

*Table A1.* Hyperparameters used for each model architecture.

| MODEL | TRAINABLE PARAMETERS | EPSILON | DELTA | LEARNING RATE | MAX GRAD NORM |
|---|---|---|---|---|---|
| ViT | ALL | 8 | $2.04e^{-5}$ | 0.00031 | 4.63 |
| RESNET | ALL | 8 | $2.04e^{-5}$ | 0.00098 | 6.53 |

### A.3. Grad sample modes in Opacus

Opacus supports multiple different gradient sampling methods as indicated in the documentation[1]. In our original experiments we used the grad_sample mode $hooks$ that is the default. This will use custom opacus modules when they are defined for that layer and functorch as a fallback. Based on the feedback by a reviewer we tried out different methods listed in the documentation for both ResNet and ViT models:

- $functorch$: We forced opacus to use functorch but did not observe any significant speed differences to using $hooks$. This is in line with the opacus documentation which writes that the speed is $0-50\%$ slower than $hooks$.

- $ExpandedWeigths$: We tried this approach but ran into runtime errors. Interestingly, when looking through the issues others have reported issues[23] but it seems to be more a PyTorch problem and has not been addressed for years. According to the documentation $ExpandedWeights$ is still in beta status.

- $GhostClipping$: Note that this method only works for ViT as described in Sec. 6.1. At first we did not manage to decrease the loss with this implementation due to the implementation in opacus being unstable. After some fixes, the correct accuracy is achieved but we noticed that the speed-ups are not significant, and even lower than flat clipping. Therefore, we decided to not include them, as it is still in development. When ready, we expect a similar speed-up to the observed in our experiments in Sec. 6.1 as the underlying algorithm is the same.

### A.4. Poisson Subsampling JAX Algorithm

We present our DP-SGD implementation in JAX that uses the correct Poisson subsampling and therefore we can account for its privacy. The main problem with implementing DP-SGD with JAX is the batches of variable size. In order to address this issue, we compute always full physical batches and mask out gradients so that the total number of used gradients is equal the sampled logical batch sizes. This means that we always compute a little more gradients that required due to sampling. This prevents the recompiling.

**Variability in experiments** One difference between the two frameworks is the variability in the experiments. PyTorch runs are remarkably consistent, maintaining low variance, and yielding the same throughput result for a fixed seed. In

---

[1]`https://github.com/pytorch/opacus/tree/61ae0ea4fb37a835e93040b5de19e8dfcd465a07/opacus/grad_sample`

[2]`https://github.com/pytorch/opacus/issues/464`

[3]`https://github.com/pytorch/opacus/issues/584`

---

**Algorithm A1** Virtual Batching DP-SGD JAX

---

**Input:** Training data points $\{x_1, \ldots, x_N\}$, loss function $\mathcal{L}(\theta) = \frac{1}{N}\sum_i \mathcal{L}(\theta, x_i)$
**Parameters:** Parameters: learning rate $\eta_t$, noise scale $\sigma$, gradient norm bound $C$, number of steps $T$, expected logical batch size $L$, physical batch size $p$.
**Start**
**for** $t \in [T]$ **do**
    $tl \sim \text{Binomial}\left(N, \frac{L}{N}\right)$ {Sample the true batch size}
    Find minimum $k \in \mathbb{N}$ such that $p \cdot k \geq tl$ {Check how many full physical batches are required}
    $m \leftarrow k \cdot p$
    $B \leftarrow \{x_{j_1}, \ldots, x_{j_m}\}$
    $P \leftarrow \{B_1, \ldots, B_k\}$ {Divide the maximum logical batch $B$ into physical batches of size $p$}.
    $M \leftarrow \{1_0, 1_1, \ldots, 1_{tl-1}, 0, 0, \ldots, 0_{m-tl+1}\}$ {Create masks so that $\sum_i^m M_i = tl$}
    $\theta_{acc} \leftarrow \mathbf{0}$
    **for** $s \in [P]$ **do**
        **for** $i \in s$ **do**
            $\mathbf{g}_t(x_i) \leftarrow \nabla_{\theta_t} \mathcal{L}(\theta_t, x_i)$ {**Compute gradient**}
            $\bar{\mathbf{g}}_t(x_i) \leftarrow M_{i+(s-1)*p} \cdot \mathbf{g}_t/\max(1, \frac{\|\mathbf{g}_t(x_i)\|_2}{C})$ {**Clip gradient and mask**}
        **end for**
        $\theta_{acc} \leftarrow \theta_{acc} + \sum_i \bar{\mathbf{g}}_t(x_i)$ {**Accumulate gradient**}
    **end for**
    $\widetilde{\mathbf{g}}_t \leftarrow \frac{1}{|L|}(\theta_{acc} + \mathcal{N}(0, \sigma^2 C^2 \mathbf{I}))$ {**Add noise**}
    $\theta_{t+1} \leftarrow \theta_t - \eta_t \widetilde{\mathbf{g}}_t$ {**Step**}
**end for**
**Return** Learned parameters $\theta_T$ and the privacy cost from a privacy accountant.

---

contrast, JAX naive executions, meaning that there is recompilation, are more variable than those of PyTorch, likely due to its sensitivity to HPC environment fluctuations and accelerator stochasticity, as noted in Fig. A.4. Additionally, JAX's asynchronous dispatch method complicates time benchmarking by issuing a promise rather than immediate results, thereby concealing Python overheads. For our Masked DP-SGD method, by fixing the batch sizes and avoiding recompilation, we achieve consistent execution times.

# B. Additional Results

This section provides additional figures that supplement the findings in the main text.

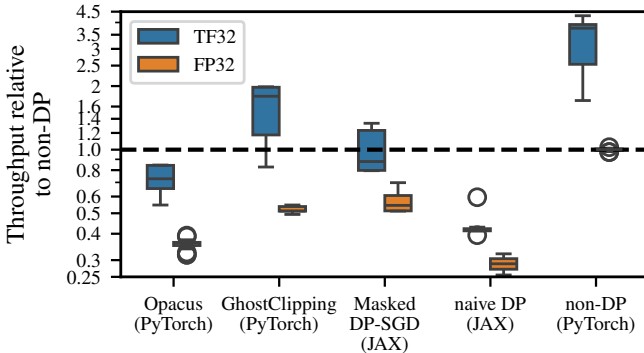

*Figure A.1.* Relative throughput to the respective FP32 non private baseline (higher is better) on NVIDIA A100. For each optimization method and each model size, we divide its throughput with the non-private counterpart. In this figure we showcase for each optimization, both precision modes, relative to the FP32 baseline.

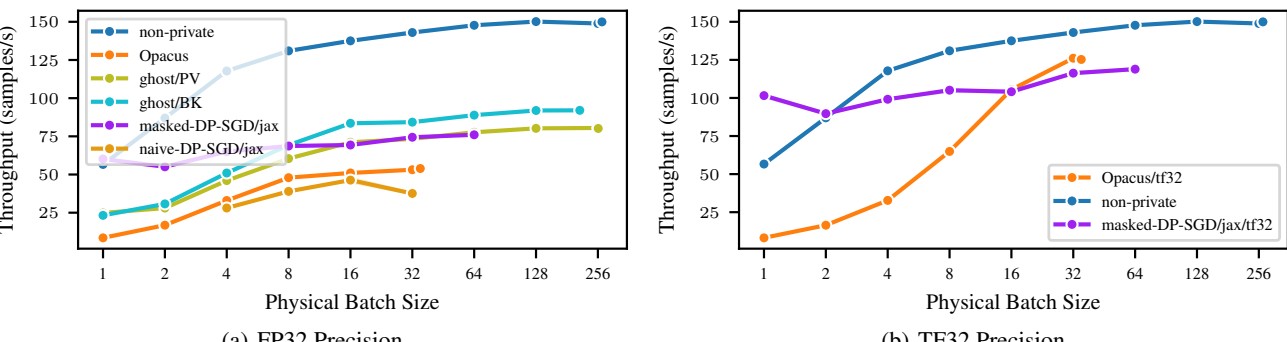

(a) FP32 Precision      (b) TF32 Precision

*Figure A.2.* Comparison of the throughput as a function of the physical batch size between the JAX and PyTorch clipping algorithms on A100 GPU. The analysis excludes the Mix algorithms, due to their equivalent performance in ViTs.

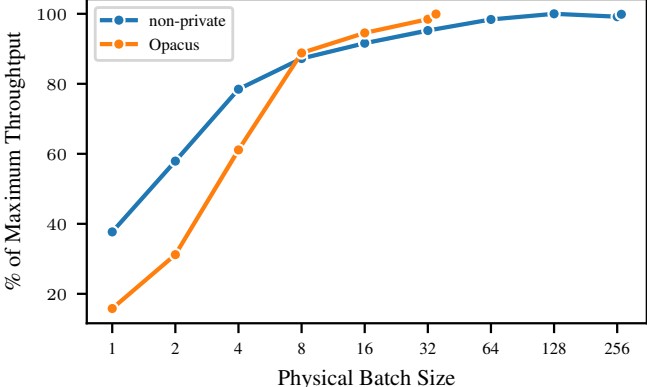

*Figure A.3.* The relative difference with the throughput at the maximum batch size for the ViT base model on A100.

*Table A2.* Mean accuracy for CIFAR-100 test set for each clipping mode for the ViT models on A100 after training for two epochs. All use the ViT hyperparameters from Table A1. While this work does not focus on the model's utility, having their results still allows us to compare them. The use of TF32 as a lower precision mode does not affect the model's utility.

| CLIPPING MODE | TEST ACCURACY |
| --- | --- |
| OPACUS | 0.8223 |
| OPACUS/TF32 | 0.8225 |
| JAX NAIVE | 0.8146 |
| MASKED DP-SGD | 0.8224 |
| PV-GHOST | 0.822 |
| BK-GHOST | 0.822 |

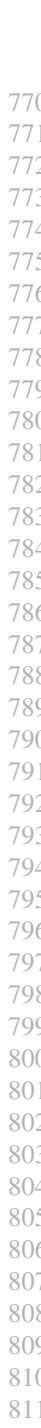

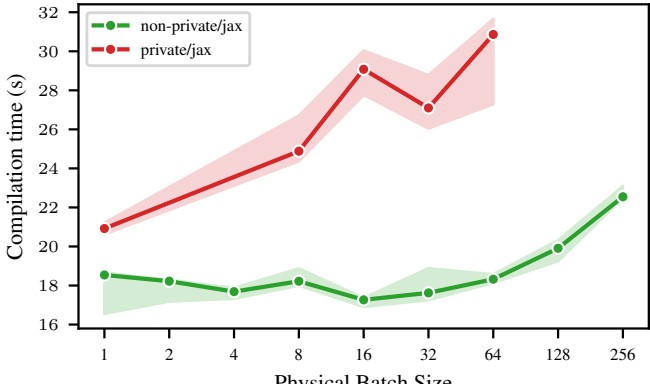

*Figure A.4.* Compilation time in seconds as a function of the physical batch size for JAX naive experiments for the ViT Base model on A100. The estimator is the median and the error bars are the 95% confidence interval using bootstrapping.

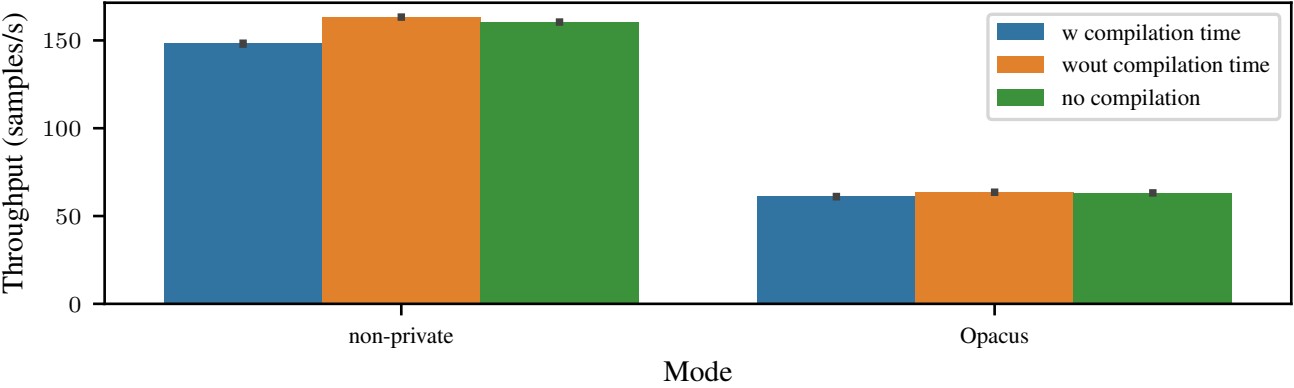

*Figure A.5.* Torch compilation experiments on A100, using the maximum physical batch size for each mode and ViT Base. PyTorch 2 enables compiling the model to (potentially) gain further speed-ups. We tried PyTorch 2 compilation to make a fair comparison with the JAX compilation but did not observe any benefits from it. We found that when trying to compile PyTorch, it first tries to compile but then falls back to NVIDIA kernels and optimizations. In the end, it does not compile, and the throughput is the same. If we take into account the first iteration (w compilation time), it is worse because of the time PyTorch spends trying to compile before falling back to NVIDIA kernels and optimizations. Disregarding the time where PyTorch tries to compile (wout compilation time), leads to nearly the same throughput as the version that does not attempt using PyTorch 2 compiling in the first place.

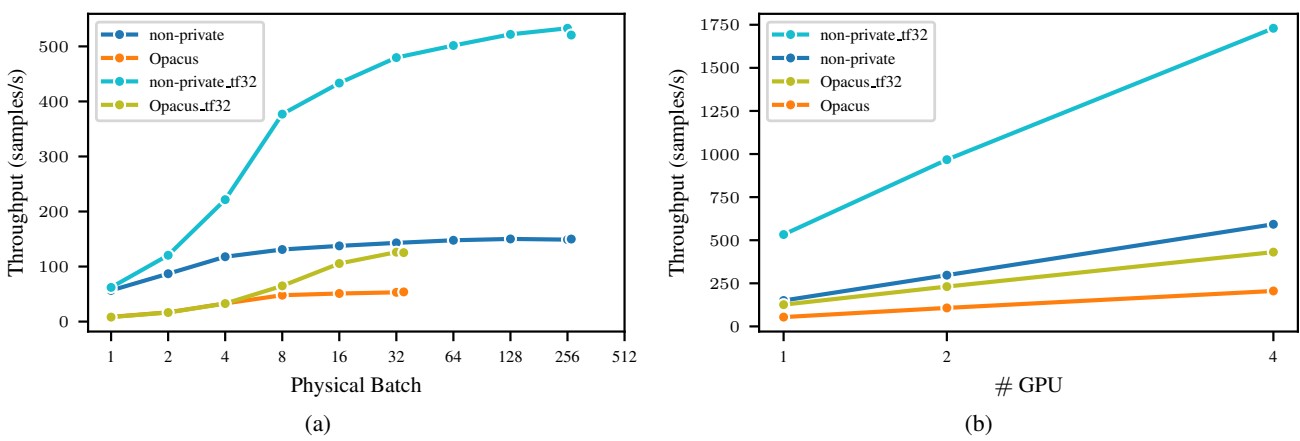

*Figure A.6.* Combining distributed training with the use of lower precision TF32 for the ViT base model on A100. (a) Throughput for one GPU; (b) Throughput for multiple GPUs.

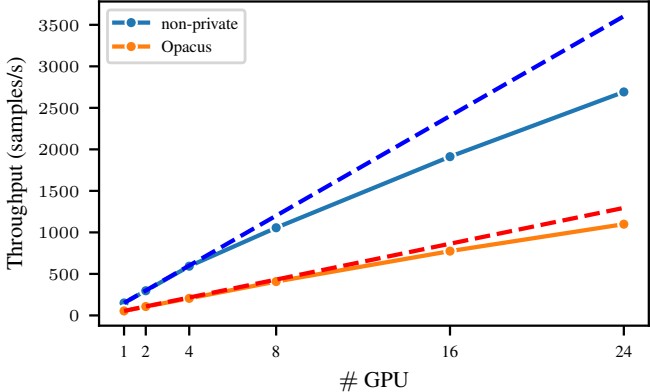

*Figure A.7.* Comparison between the throughput by scaling the number of GPUs with more nodes for the non-private and Opacus training with the ViT base model on A100 GPUs. The dashed line is the ideal growth if it were linear.

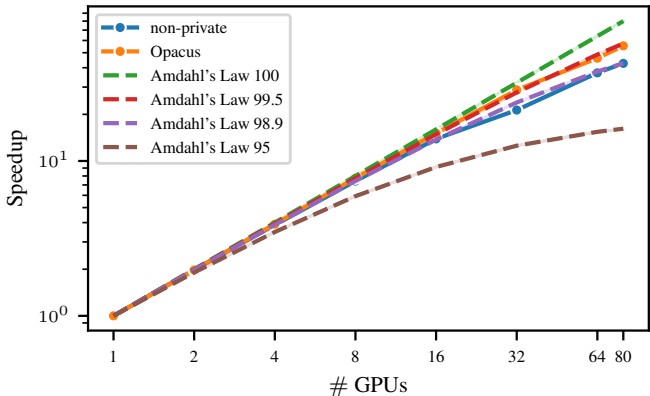

*Figure A.8.* Comparison between the throughput in our experiments and the theoretical Amdahl's Law. Both axis are in log scale. In the distributed setting, private training achieves a 99.5 % of parallel processing, with a 50 times speed up than single processing.

## C. Further discussion of TF32 speedups

The speedup observed in Fig. 5 peaks at the "base" model. We believe that the reasons are the following: Speed-ups resulting from TF32 can significantly vary on per case basis as "all storage in memory and other operations remain completely in FP32, only convolutions and matrix-multiplications convert their inputs to TF32 right before multiplication." (Stosic & Micikevicius, 2021). Until now, TF32 precision benchmarks have been limited to non-DP applications which was one of the reasons we wanted to discuss our observations in DP context. It appears the effectiveness of TF32 arithmetic peaks at "base" configuration. This due to a mix of reasons which are difficult to quantify exactly. Firstly, it is likely that matrix multiplication kernel dominance peaks at this configuration i.e. we have the most parameters whilst the batch size dimension also remains sufficiently large. With large and huge model variants the parameter count still increases but at the cost having very small batch dimension of 10 and 3, respectively. Secondly, we observe similar trend in Fig. 2(a) where the discrepancy between dp and non-dp grows as model size gets bigger. This suggests that the dominance of DP operations also grows with the model size. None of the DP-operations are cast as matrix-multiplications and hence won't benefit from TF32.

## D. Extra computational cost of the `masked dp-sgd`

For the `masked dp-sgd`, we first sample the minibatch using Poisson subsampling and to allow JAX compilation, we round this number to the closest larger integer divisable by the physical batch size. Hence, for any sampled batch size $X$, the difference between $X$ and the upscaled batch size will be in $\{0, \ldots p-1\}$ for a physical batch size $p$. Denoting the excess batch size with $\Delta_p(X)$ and the upscaled batch size with $X_+$, we can write

$$\mathbb{E}[X_+] = \mathbb{E}[X + \Delta_p(X)]. \tag{A1}$$

Now, we can form a simple upper bound for the expected value of the upscaled batch size as

$$\mathbb{E}[X_+] \leq \mathbb{E}[X] + (p-1). \tag{A2}$$

When working large number of samples and non-negligible sampling probabilities, the excess cost due to upscaling the batch size will be modest for feasible physical batch sizes. For example, in our experiments the expected batch size of the Poisson subsampling was $25\,000$, whereas the physical batch sizes extended up to $64$.

A recent work by (Chua et al., 2024b) proposed an alternative implementation for JAX compilable implementation of Poisson subsampled DP-SGD. In their approach the batch sizes are sampled from a truncated Binomial distribution. This affects the privacy guarantees of the models, and therefore they need to compensate the truncated sampling by increasing the noise std. for DP-SGD. They suggest an approach for computing the truncation bound $B$ as

$$\Psi(n, b, B) \cdot T \cdot (1 + e^\epsilon) \leq \tau\delta \tag{A3}$$

where $\Psi(n, b, B)$ denotes the survival function $(1 - \texttt{cdf})$ of $\text{Binom}(n, b/n)$ at $B$ and $T$ are the number of steps. The parameter $\tau$ effectively scales the size of the tails and is used to calibrate the noise std by selecting $\sigma$ such that the hockey-stick divergence between the Poisson subsampled Gaussian mechanisms is bound by $(1 - \tau)\delta$. (Chua et al., 2024b) choose $\tau = 10^{-5}$, which keeps the noise std. increase very small.

In the implementation of (Chua et al., 2024b), the gradients are computed for $B$ randomly selected samples, after which the final samples are chosen according to the batch size sampled from the truncated Binomial. Hence the computational excess over regular Poisson subsampling becomes $B - b$. For example, in our setting where $\epsilon = 8$, $\delta = 10^{-5}$, $n = 50\,000$, $b/n = 1/2$ and $T = 4$, the $B - b = 858$, which is significantly larger than the $p - 1$ excess of our method for obtainable physical batch sizes ($p \leq 64$).

