# OpenReview forum: "Towards Efficient and Scalable Implementation of Differentially Private Deep Learning"
_ICML.cc/2025/Conference — Submitted to ICML 2025_

### Official Review · Reviewer_rRoP · 2025-03-07

**Overall Recommendation:** 4

**Summary:**

This focus of the paper is on computational efficiency of implementing Differentially Private Stochastic Gradient Descent (DP-SGD), which is commonly used for private ML model training. In particular, the paper focuses on
1. _Poisson subsampling for generating batches_: while typical implementation have used shuffling based batches, recent work has shown that this can have a worse privacy guarantee.
2. _JAX based implementation_: The paper reports that non-private SGD implemented in JAX can be faster than PyTorch, however a naive implementation of DP-SGD in JAX suffers a much lower throughput due to recompilation of computation graphs in JAX.

The contributions of the paper include:
*  It proposes a novel JAX-based implementation of DP-SGD called _Masked DP-SGD_, which correctly implements Poisson subsampling while avoiding the JAX recompilation issues, that a naive implementation would suffer due to variable batch sizes. The idea is to use the standard approach of using small physical batches and gradient accumulation to simulate large logical batches. But to adapt this for Poisson subsampling where the batch size can be varying, some of the examples are masked out for each optimization step.
* The paper evaluates and benchmarks various strategies for reducing the computational cost of DP-SGD. This includes efficient gradient clipping techniques like Ghost Clipping, Mixed Ghost Clipping, and Book Keeping. It is reported that efficient clipping implementations can roughly halve the cost compared to a naive implementation in Opacus (currently supported in PyTorch). Th JAX implementation of Masked DP-SGD achieves comparable or better performance than efficient PyTorch-based methods.
* It studies the impact of lower precision training using TF32 (representation) on the throughput of DP-SGD, finding potential speedups for certain model sizes. The paper also notes concerns regarding the theoretical privacy guarantees in such settings.
*  It also studies the scalability of DP-SGD in distributed training environments (with multiple GPUs), demonstrating that DP-SGD scales even better than non-private SGD when using a large number of GPUs (up to 80), likely due to its slower pace and less frequent network saturation.

### Post-rebuttal update
I continue to maintain my score and recommendation.

**Claims And Evidence:**

All claims made in the paper are supported by evidence. It is great that the entire source code for experimentation is made available.

**Essential References Not Discussed:**

As far as I can tell, all relevant literature is adequately discussed and cited.

**Experimental Designs Or Analyses:**

The experimental setup looks sound to me.

**Methods And Evaluation Criteria:**

The proposed method and evaluation criteria are sound.

**Other Comments Or Suggestions:**

### Minor comments:
* Figure 5: The y-axis is not clear. Is it the ratio of throughput of TF32 / FP32 or FP32 / TF32? I think it is the former, but it would be better to be explicit about the same.

**Other Strengths And Weaknesses:**

### Strengths

* _Thorough Empirical Evaluation:_ The paper presents extensive benchmarking results across different model architectures (Vision Transformers and ResNets), frameworks (PyTorch and JAX), and hardware (NVIDIA V100 and A100 GPUs).
* _Novel JAX Implementation:_ The proposed Masked DP-SGD method offers a promising avenue for efficient and correct DP-SGD training, overcoming the recompilation issues when naively implementing in JAX.
* _Open Source Contribution:_ The source code for their implementation is made available in supplementary material, which I highly appreciate!

Overall, this paper makes a significant contribution to the field by providing a detailed empirical analysis of the computational costs associated with correctly implemented DP-SGD and by proposing and evaluating effective optimization strategies, including a novel JAX-based approach. The findings are relevant to both researchers and practitioners working on deploying differentially private deep learning.
I recommend acceptance.

**Questions For Authors:**

One thing that was not clear to me in _Masked DP-SGD_ is how the batches are sampled at each step. The pseudocode simply says $B \\gets \\{x_{j_1}, \\ldots, x_{j_m}\\}$, but how to do this efficiently is not clear in cases where the dataset is too large to fit in memory. Perhaps _Masked DP-SGD_ can be applied in conjunction with the method of _Scalable DP-SGD_ proposed in [Chua et al. (2024b)](https://www.arxiv.org/abs/2411.04205), which is supposed to work when the dataset size is very large?
I would appreciate some discussion about the same.

**Relation To Broader Scientific Literature:**

The paper positions itself nicely within existing literature by providing an exhaustive empirical evaluation of different methods for implementing DP-SGD. This is helpful even beyond the introduction of the "Masked DP-SGD" technique that it introduces.

**Theoretical Claims:**

There aren't any non-trivial theoretical claims in the paper.

---

> ### Author Rebuttal · Authors · 2025-03-31
>
> Thanks for your time and your insightful review.
>
> > Figure 5:
>
> Thanks for the comments! It is correct, the ratio of the throughput is TF32/FP32 and that should be clarified in the figure caption. It will be updated for the camera ready version.
>
>
> > One thing that was not clear to me in Masked DP-SGD is how the batches are sampled at each step.
>
> The masked DP-SGD batch sampling can be composed in two main steps. First, we sample the batch size $B^{(t)}$ from a Binomial distribution $\text{Bin}(N, q)$, where N is the number of samples and $q$ is the subsampling rate. We round $B^{(t)}$ up to the next integer multiple of physical batch size $p$: $B_+^{(t)} = p\lceil B^{(t)} / p\rceil$. Second we permute the data set, and pick the first $B_+^{(t)}$ elements as the full logical batch that we process as chunks of $p$ samples. Finally when the gradients are aggregated, we throw away (by zeroing the gradients) for the padded $B_+^{(t)} - B^{(t)}$ samples.
>
>
> > How to do this efficiently is not clear in cases where the dataset is too large to fit in memory. Perhaps Masked DP-SGD can be applied in conjunction with the method of Scalable DP-SGD proposed in Chua et al. (2024b), which is supposed to work when the dataset size is very large?
>
> This is a very interesting future direction that we are happy to discuss! Indeed the current sampling procedure of the Masked DP-SGD might struggle with very large data sets, as permuting the indices becomes very expensive in that case. We could indeed combine the Chua et al. (2024b) approach for the subsampling with the masked approach by simply replacing the Truncate/Pad step in their approach with our masking step, and processing the minibatches again as chunks of $p$.

---

> > ### Comment · Reviewer_rRoP · 2025-04-02
> >
> > Thanks for the response. Actually now I am even more confused about how the batches are sampled in Masked DP-SGD. I understand the part about sampling the batch size $B^{(t)}$ from the Binomial distribution, but the part about permuting the dataset seems inefficient. Doesn't that require a random permutation at _each_ step of training? That seems quite inefficient to me... If the implementation is actually permuting the dataset once, then that would be an incorrect implementation of Poisson subsampling.

---

> > > ### Author Response · Authors · 2025-04-03
> > >
> > > Thanks for your reply and your very detailed question regarding our implementation!
> > >
> > > > If the implementation is actually permuting the dataset once, then that would be an incorrect implementation of Poisson subsampling.
> > >
> > > The reviewer is absolutely correct that the permutation needs to happen at every step to be DP. We would like to clarify that the implementation permutes at every step of Poisson subsampling (see lines 339-354 of `jax_exp/jax_mask_efficient.py` in the supplement) and thus the implementation is doing correct Poisson subsampling and is DP.
> > >
> > > Lines 339-354 of `jax_exp/jax_mask_efficient.py`:
> > > ```
> > > for t in range(num_iter):
> > >     	sampling_rng = jax.random.PRNGKey(t + 1)
> > >     	batch_rng, binomial_rng, noise_rng = jax.random.split(sampling_rng, 3)
> > >
> > >     	#######
> > >     	# poisson subsample
> > >     	actual_batch_size = jax.device_put(
> > >         	jax.random.bernoulli(binomial_rng, shape=(full_data_size,), p=q).sum(),
> > >         	jax.devices("cpu")[0],
> > >     	)
> > >     	n_physical_batches = actual_batch_size // physical_bs + 1
> > >     	logical_batch_size = n_physical_batches * physical_bs
> > >     	n_masked_elements = logical_batch_size - actual_batch_size
> > >
> > >     	# take the logical batch
> > >     	indices = jax.random.permutation(batch_rng, full_data_size)[:logical_batch_size]
> > > ```
> > >
> > >
> > > > I understand the part about sampling the batch size from the Binomial distribution, but the part about permuting the dataset seems inefficient. Doesn't that require a random permutation at each step of training?
> > >
> > > Apologies, we believe that we might have not carefully formulated our reply and thus caused a misunderstanding. We wrote that we “permute the data set, and pick the first $B_+^{(t)}$ elements as the full logical batch” while in the implementation we work on the indices instead of the dataset itself (see part of supplement code above).
> > >
> > > > That seems quite inefficient to me...
> > >
> > > Thanks for pointing this out!
> > >
> > > The reviewer is correct that permuting the indices of the dataset at each iteration can be costly, especially with larger datasets it becomes significantly more expensive than the uniform sampling done in e.g. [Opacus](https://github.com/pytorch/opacus/blob/6c2cde9cc715f6c45983901461e06d9abad09fea/opacus/utils/uniform_sampler.py#L150-L158). Fortunately, we can actually easily adapt the Opacus style sampling to our Masked-SGD implementation!
> > > 1. We first sample the (actual) logical batch using Poisson subsampling the same as Opacus.
> > > 2. Next, we pad the batch with arbitrary elements to make its size an integer multiple of the physical batch size.
> > > 3. Finally, as we do with our original implementation, we mask away the padded samples.
> > >
> > > As the masking removes the effect of the padded samples, we could for example repeat elements of the sampled batch for the padding (essentially wrapping around the indices), or we could pad with the first $B_+^{(t)} - B^{(t)}$ elements of the full dataset. As the padding is a constant time operation, the complexity of this proposed sampling procedure would match that of Opacus.
> > >
> > > We implemented this variant below and profiled it with different `full_data_size` and found that indeed the new implementation outperforms the old version when `full_data_size` is sufficiently large.
> > >
> > > The below table shows the average number of seconds it takes for sampling the methods as a function of `full_data_size` when executing it on the CPU of the cluster we used for our experiments. We average over 10 repeats when discarding the initial compiling time for both. (We used `block_until_ready` to profile the function executions).
> > >
> > > | sampling_method   |	n=10 000 |n=100 000 |n=1 000 000 |n=10 000 000
> > >  |:----------------|---------:|----------:|----------:|-----------:|
> > > | old       	| 0.033 | 0.058 |  0.645 |  11.353	|
> > > | new      	| 0.150 | 0.157  |  0.229 |   0.623 |
> > >
> > >
> > >
> > > This source code is the new sampling method that is inspired by discussion that the reviewer initiated.
> > >
> > > ```
> > > def sample_batch_new_version(seed, full_data_size):
> > > 	sampling_rng = jax.random.PRNGKey(seed)
> > > 	batch_rng, binomial_rng, noise_rng = jax.random.split(sampling_rng, 3)
> > >
> > > 	#######
> > > 	# poisson subsample
> > > 	poisson_subsampled_indices = jax.random.bernoulli(batch_rng, p=q, shape=(full_data_size,)).nonzero()[0]
> > > 	actual_batch_size = len(poisson_subsampled_indices)
> > >
> > > 	n_physical_batches = actual_batch_size // physical_bs + 1
> > > 	logical_batch_size = n_physical_batches * physical_bs
> > > 	n_masked_elements = logical_batch_size - actual_batch_size
> > >
> > > 	# take the logical batch
> > > 	pad = poisson_subsampled_indices[:n_masked_elements]
> > > 	indices = jnp.concatenate([poisson_subsampled_indices, pad])
> > >
> > > 	masks = jax.device_put(
> > >     	jnp.concatenate([jnp.ones(actual_batch_size), jnp.zeros(n_masked_elements)]),
> > >     	jax.devices("cpu")[0],
> > > 	)
> > >
> > > 	return indices, masks
> > > ```

---

### Official Review · Reviewer_1fg2 · 2025-03-14

**Overall Recommendation:** 1

**Summary:**

The paper provides a comprehensive empirical study of Differentially Private Stochastic Gradient Descent (DP-SGD) implementations that properly incorporate Poisson subsampling, which is crucial for maintaining theoretical privacy guarantees. Recent research has demonstrated that many implementations ignore the Poisson subsampling requirement, potentially compromising privacy guarantees.

The authors benchmark existing PyTorch and JAX implementations, as well as introducing their own JAX implementation with proper Poisson sampling. They also propose "Masked DP-SGD," a novel approach that avoids expensive recompilation in JAX, leading to substantial efficiency gains.

The authors provide practical recommendations for more efficient DP-SGD deployments. Their findings include insights on gradient clipping optimizations, precision modes, and scaling behavior in distributed settings.

## Post-rebuttal update.

After reading the rebuttals, I choose to maintain my score, albeit with a lower confidence (as I've indicated in the AC discussion).
I'm highly confident in my assessment of the technical side of the paper - it is very strong and thorough.
I only have medium-level confidence in my scope assessment. My score reflects my intuition based on my current understanding of ICML expected scope and paper's contribution, but I'm open to reconsider if the consensus between AC and other reviewers disagrees with my assessment.

**Claims And Evidence:**

The paper provides a comprehensive set of benchmarks for both throughput and memory consumption across a wide range of widely adopted state-of-the-art DP-SGD implementations, covering both PyTorch and JAX frameworks.

All findings are well-documented with necessary implementation details, allowing for reproducibility and clear understanding of the experimental setup. The benchmarks use consistent metrics and evaluation criteria across implementations.

I believe, however, that the scope of the paper is not broad enough to be considered for ICML. While it's a very solid technical work and important for practitioners, the paper lacks scientific novelty that would be expected at ICML.

The proposed method (Masked DP-SGD) represents an incremental technical improvement that is highly specific to the JAX framework. While valuable for JAX users, it doesn't present a fundamental advancement in the field of differentially private machine learning that generalizes beyond this specific implementation context.

**Essential References Not Discussed:**

None

**Experimental Designs Or Analyses:**

See above

**Methods And Evaluation Criteria:**

Paper's methodology is solid and covers a good range of models and datasets in realistic evaluation scenarios. The authors examine both Vision Transformer (ViT) and ResNet architectures of varying sizes, providing a comprehensive view of how different implementations perform across model scales.

Looking through the appendix, it's clear the authors took great care to extensively evaluate existing methods, including obscure implementation details like "grad_sample_mode" in Opacus.

The authors also employed NVIDIA profiling tools for deeper insights into computational bottlenecks, allowing them to identify specific causes of performance differences between implementations. This profiling helps explain why certain optimizations are effective and provides valuable information for practitioners.

**Other Comments Or Suggestions:**

N/A

**Other Strengths And Weaknesses:**

N/A

**Questions For Authors:**

N/A

**Relation To Broader Scientific Literature:**

The paper fits well with recent attention on the important question of DP-SGD applications with proper Poisson sampling. It addresses concerns raised by works like Lebeda et al. (2024), Chua et al. (2024a/b), and Annamalai et al. (2024), which highlight that many implementations have weaker privacy guarantees than claimed due to improper sampling.

Compared to previous papers with DP-SGD benchmarks, this one focuses extensively on proper Poisson sampling, filling an important gap in the literature. While earlier works compared efficiency of different implementations, they often overlooked the sampling requirement that is crucial for theoretical privacy guarantees.

I believe, however, that this paper lacks a substantial novel contribution to the field, as it mostly focuses on technical details of existing implementations. The proposed Masked DP-SGD method, while useful, represents an engineering solution to a framework-specific problem rather than advancing our understanding of differentially private learning more broadly.

**Theoretical Claims:**

N/A

---

> ### Author Rebuttal · Authors · 2025-03-31
>
> Thanks for your time and your careful review.
>
> We respectfully disagree with your statement that “While it's a very solid technical work and important for practitioners, the paper lacks scientific novelty that would be expected at ICML.” because the call for papers of ICML mentions as topic of interest: “Machine Learning Systems (improved implementation and scalability, hardware, libraries, distributed methods, etc.)”. We believe that our work fits in the call for papers.
>
> ICML Call for Papers has a section for review criteria, which starts with: “Submissions should report original and rigorous research of significant interest to the machine learning community.” We believe that according to your review, our paper satisfies this completely, when taking into account that the *machine learning community* includes both the researchers and practitioners.

---

### Official Review · Reviewer_NagC · 2025-03-14

**Overall Recommendation:** 3

**Summary:**

This paper investigates the computational efficiency of DP-SGD, with a focus on analyzing the computational cost of using Poisson subsampling for DP training, and comparing a series of DPSGD schemes. To reduce computational costs, the author proposed Masked DPSGD algorithm by addressing the frequent recompilation problem caused by Poisson sampling. The study also explored the application of low precision computation (TF32) in DPSGD and found that it can improve computational throughput, but its impact on privacy guarantee still needs further research.

**Claims And Evidence:**

Yes. Most of the claims in the paper are well supported by the experimental results.

**Essential References Not Discussed:**

The reviewer thinks the references are appropriate.

**Experimental Designs Or Analyses:**

I've checked the experiments on the computational overhead of DP-SGD, covering core results such as throughput and batch size of different DPSGD methods. No significant issues were spotted.

**Methods And Evaluation Criteria:**

Metrics: Yes. Throughputs and maximum achievable physical batch size are used as metrics to evaluate computational overhead.

Datasets: No. Only CIFAR100 the image dataset is used.

Models: Probably okay. ViT and ResNet families are adopted. Perhaps language models could be considered as well.

**Other Comments Or Suggestions:**

On the right-bottom of page 2: 'using the the per-example ...'

The authors could further enhance their writing. For example, the main algorithm should be put in the main text but not the appendix.

**Other Strengths And Weaknesses:**

Strengths: the paper views DPSGD from a computational efficiency perspective which is meaningful and interesting. It corrects the Poisson Subsampling implementation issues in previous works and proposes its own method with a higher efficiency.

Weaknesses: the privacy guarantee of the proposed algorithm is missing. Lower precision using TF32 increasing the throughput is interesting but may not ensure its good performance on other tasks. The experimental results are limited to one single image dataset. Since the privacy exhibits a tradeoff with accuracy, it may be beneficial to discuss accuracy in the experiments.

**Questions For Authors:**

1. What is the privacy guarantee of your proposed algorithm? How can you prove it?
2. What is the computational overhead of these DPSGD methods on datasets beyond CIFAR100?
3. Why is accuracy missing from all experimental results? How does the sampling affect accuracy results?

**Relation To Broader Scientific Literature:**

The work re-implemented several DP-SGD methods, such as opacus, ghost clipping, mixed ghost clipping, Book Keeping ghost, etc., with Poisson subsampling and compared their computational overheads.

The work proposed a new implementation which has a superior throughput than the aforementioned methods.

**Theoretical Claims:**

I have checked Alg. A1: Virtual Batching DP-SGD JAX; unfortunately, the paper seems to miss an important theorem on privacy guarantee that Alg. A1 satisfies.

---

> ### Author Rebuttal · Authors · 2025-03-31
>
> Thanks for your time and your thorough review.
>
> > What is the privacy guarantee of your proposed algorithm? How can you prove it?
>
> The DP guarantees of the **Masked DP-SGD** follow from the standard analysis of Poisson subsampled DP-SGD in the add/remove adjacency. Note that the only difference we have to standard Poisson subsampling, is that in cases where the sampled batch size $B$ is not an integer multiple of the physical batch size $p$, we need to pad the batch with additional samples. We compute the clipped per-example gradients for these $B_+ = p \lceil B / p  \rceil$ samples. However, when aggregating the clipped per-example gradients in a sum, we weigh the padded $B_+ - B$ samples with $0$. Hence, the sampling procedure we use is equivalent to that of Poisson subsampling, allowing us to analyze the privacy guarantees as the Poisson subsampled Gaussian mechanism.
>
> > What is the computational overhead of these DPSGD methods on datasets beyond CIFAR100?
>
> This is an interesting question. In this work we focussed on cases where the complete dataset can be still handled by one machine and in these cases the computational overhead is mostly influenced by the model (see Figures 1 and 2). Handling cases where distributed computing is needed might require additional methodology that is orthogonal to our advancements (see also discussion with reviewer rRoP).
>
> > Lower precision using TF32 increasing the throughput is interesting but may not ensure its good performance on other tasks. [...] Since the privacy exhibits a tradeoff with accuracy, it may be beneficial to discuss accuracy in the experiments.
>
> Thanks for the suggestion, we added new experiments below complementing the Table A2 (in the Appendix), where we compare accuracy for the ViT base model on CIFAR100. The new experiments measure the accuracy and throughput for the SVHN and CIFAR10 datasets with different numbers of examples per class and hyperparameters. We are comparing the accuracy and throughput between precision modes for private training with Opacus.
>
> | dataset | epochs | S   | lr       | throughput FP32 | throughput TF32 | accuracy FP32 | accuracy TF32 | std FP32 | std TF32 |
> |---------|-------|-----|----------|-----------|----------|------------|----------|------------|----------|
> | cifar10 | 18    | 250 | 0.000710 | 57.317226 | 117.987103 | 0.919950  | 0.923300 | 0.008273   | 0.008485 |
> |         | 6     | 100 | 0.000758 | 56.950525 | 114.012746 | 0.956275 | 0.956367 | 0.001504 | 0.001850 |
> | SVHN    | 23    | 250 | 0.00098  | 57.279881 | 117.286507 | 0.610933 | 0.611321 | 0.009724 | 0.009848 |
> |         | 23    | 500 | 0.00098  | 57.352352 | 118.368599 | 0.806301 | 0.806438 | 0.006607 | 0.007316 |
>
>
> The throughput difference is the same for both datasets. Using TF32 is twice as fast as FP32. We tested the variability in accuracies across three repeats of the DP training with both TF32 and FP32. We computed the pairwise differences between TF32 and FP32 on different seeds and data sets. The differences between the two precisions have a mean of $\approx -4.3 \times 10^{-5}$ and std $\approx 4.2 \times 10^{-4}$. Using a pairwise t-test on the differences we cannot reject the null hypothesis that the accuracies have the same mean (p-value $\approx 0.74$). Furthermore, compared to the variance arising from DP (see Table above), the differences from changing the precision are negligible. This is in line with the previous results of the article in Table A2.
>
> > Why is accuracy missing from all experimental results? How does the sampling affect accuracy results?
>
> The paper focuses on computational efficiency and as can be seen from Table A2 the optimizations have no impact (apart from tiny impact due to seeding noise) on test accuracy. We use the accuracy to test our implementations and check that everything is implemented correctly as no accuracy difference is expected unless using different precision where we observe little impact (see Table A2 and experiments above).
>
> The general question on which sampling is optimal is a separate question and there is some early work looking at this question [1] but for fair comparisons between sampling in terms of accuracy the accounting must match the sampling method and so far tight accounting has been only established for Poisson subsampling [2]. Tight accounting for other sampling methods remains an active area of research.
>
> [1] Chua et al. Scalable DP-SGD: Shuffling vs. poisson subsampling. NeurIPS 2024.
>
> [2] Annamalai et al.. (2024) To Shuffle or not to Shuffle: Auditing DP-SGD with Shuffling. arXiv:2411.10614.

---

> > ### Comment · Reviewer_NagC · 2025-04-05
> >
> > The reviewer thanks the authors for addressing its concerns, as most of them are well explained. The reviewer has the remaining suggestion:
> >
> > The reviewer understands that the current experiments focus on the setting of CIFAR100 and a single machine. But the reviewer still hopes to see how the method performs on larger datasets in the future. This may make the proposed method more useful and general.
> >
> > Thanks for the comparison between TF32 and FP32. But the reviewer noticed that TF32 gives even higher accuracies than FP32, which is a bit strange. Normally, one would consider that higher precision like FP32 should give better accuracy. Maybe the author can explain a little why this happens, even if the difference is not significant.

---

> > > ### Author Response · Authors · 2025-04-09
> > >
> > > > Thanks for the comparison between TF32 and FP32. But the reviewer noticed that TF32 gives even higher accuracies than FP32, which is a bit strange. Normally, one would consider that higher precision like FP32 should give better accuracy. Maybe the author can explain a little why this happens, even if the difference is not significant.
> > >
> > > Thanks for the question regarding the utility difference between TF32 and FP32. As the reviewer has pointed out, the difference that we observe is not statistically significant and we would like to point to Stosic and Micikevicius [1] that report similar observations in the non-DP setting: the accuracies can change either up or down, but the differences are very small.
> > >
> > > TF32 only modifies certain operations as noted by [1]: *“TF32 is only exposed as a Tensor Core operation mode, not a type. All storage in memory and other operations remain completely in FP32, only convolutions and matrix-multiplications convert their inputs to TF32 right before multiplication.”*
> > >
> > > The change from FP32 to TF32 is expected to contribute slightly different rounding errors in matrix multiplications and convolutions. While this could be expected to lead to higher training loss, it is not as clear what the impact on test accuracy would be. In any case the differences are much smaller than differences caused by other sources of randomness such as different random seeds for DP.
> > >
> > > [1] Stosic, D. and Micikevicius, P. Accelerating AI training with NVIDIA TF32 tensor cores. https://developer.nvidia.com/blog/accelerating-ai-training-with-tf32-tensor-cores/, 2021

---

### Decision · Program_Chairs · 2025-05-01

**Decision:**

Reject

**Comment:**

This paper studies computational efficient implementation of DP-SGD, with a focus on Poisson subsampling for generating batches. It provided techniques to address the efficiency challenges arised to implement Poisson subsampling and evaluated the methods with extensive comparison to other methods and implementations. The paper also open sourced the implementation (included in the supplementary materials). However, there are some shared concerns among the reviewers regarding the limited novelty in the low level and narrow scoped (currently specific to JAX) techniques to improve the efficiency in processing variable batch sizes, which may not be the best suited for the ICML audience of community.